Optimizing colorectal cancer segmentation with MobileViT-UNet and multi-criteria decision analysis

Barua Barun 1
Chyrmang Genevieve 1
Bora Kangkana kangkana.bora@cottonuniversity.ac.in 1
Saikia Manob Jyoti manob.saikia@unf.edu 2 3
1 Department of Computer Science and Information Technology, Cotton University , Guwahati , Assam , India
2 Electrical and Computer Engineering Department, University of Memphis , Memphis , TN , United States of America
3 Biomedical Sensors & Systems Lab, University of Memphis , Memphis , TN , United States of America
Coelho Paulo Jorge
Electronic publication date: 2024 Dec 23
Publication date: 2024
Volume: 10
Electronic Location ID: e2633
Received 2024 Aug 7; Accepted 2024 Dec 5
Copyright: ©2024 Barua et al.
Copyright year: 2024
Copyright holder: Barua et al.
License: This is an open access article distributed under the terms of the Creative Commons Attribution License, which permits unrestricted use, distribution, reproduction and adaptation in any medium and for any purpose provided that it is properly attributed. For attribution, the original author(s), title, publication source (PeerJ Computer Science) and either DOI or URL of the article must be cited.
License URL: https://creativecommons.org/licenses/by/4.0/

Keywords: Colorectal cancer, Deep learning, Histopathology, Image segmentation, Multi decision criteria analysis

Funding: The authors received no funding for this work.

==============================
Colorectal cancer represents a significant health challenge as one of the deadliest forms of malignancy. Manual examination methods are subjective, leading to inconsistent interpretations among different examiners and compromising reliability. Additionally, process is time-consuming and labor-intensive, necessitating the development of computer-aided diagnostic systems. This study investigates the segmentation of colorectal cancer regions of normal tissue, polyps, high-grade intraepithelial neoplasia, low-grade intraepithelial neoplasia, adenocarcinoma, and serrated Adenoma, using proposed segmentation models: VGG16-UNet, ResNet50-UNet, MobileNet-UNet, and MobileViT-UNet. This is the first study to integrate MobileViT as a UNet encoder. Each model was trained with two distinct loss functions, binary cross-entropy and dice loss, and evaluated using metrics including Dice ratio, Jaccard index, precision, and recall. The MobileViT-UNet+Dice loss emerged as the leading model in colorectal histopathology segmentation, consistently achieving high scores across all evaluation metrics. Specifically, it achieved a Dice ratio of 0.944 ± 0.030 and a Jaccard index of 0.897 ± 0.049, with precision at 0.955 ± 0.046 and Recall at 0.939 ± 0.038 across all classes. To further obtain the best performing model, we employed multi-criteria decision analysis (MCDA) using the Technique for Order of Preference by Similarity to Ideal Solution (TOPSIS). This analysis revealed that the MobileViT-UNet+Dice model achieved the highest TOPSIS scores of 1, thereby attaining the highest ranking among all models. Our comparative analysis includes benchmarking with existing works, the results highlight that our best-performing model (MobileViT-UNet+Dice) significantly outperforms existing models, showcasing its potential to enhance the accuracy and efficiency of colorectal cancer segmentation.

Introduction

Colorectal cancer is the third most prevalent cancer globally, accounting for approximately 10% of all cancer cases and being the second leading cause of cancer-related deaths worldwide. It predominantly affects the elderly, with most diagnoses occurring in individuals aged 50 and older (World Health Organization, 2024). Colorectal cancer ranks as the fourth most common cancer in men and the third most common in women (Sung et al., 2021). On average, the lifetime risk of developing colorectal cancer is about 1 in 23 for men and one in 26 for women. However, individual risks can vary based on personal susceptibility factors (American Cancer Society, 2024). Colorectal cancer can originate in the colon, where it is referred to as colon cancer, or in the rectum, known as rectal cancer. Despite their different locations, these cancers are collectively termed colorectal cancer due to their similar characteristics (Pamudurthy, Lodhia & Konda, 2020; Lee et al., 2013). Often diagnosed at advanced stages, colorectal cancer presents limited treatment options (World Health Organization, 2024). Therefore, healthcare professionals emphasize the importance of regular screening, especially for individuals at higher risk due to factors such as family history or age. Routine screenings, including colonoscopy, sigmoidoscopy, and stool tests, are crucial for the early detection of colorectal cancer, which significantly enhances treatment effectiveness and prognosis. Timely identification and intervention improves the chances of successful treatment and long-term survival.

Histopathological examination of the intestinal tract remains the gold standard for diagnosing colorectal cancer and is essential for effective treatment (Thijs et al., 1996). Histopathology analysis involves the study of pathological features under the microscope by pathologists for the purpose of grading, diagnosing, and predicting the prognosis of a disease. The advantage of using intestinal biopsy for histopathological analysis lies in its ability to accurately determine the patient’s condition while causing minimal harm and promoting rapid wound healing (Labianca et al., 2013). After obtaining a biopsy, the tissue is sectioned and stained with two common staining reagents hematoxylin and eosin (H&E), this staining reagents highlights the distinctions between the nucleus and cytoplasm and revealing fine structures in the tissue (Fischer et al., 2008; Chan, 2014). When a pathologist examines the colon, they first evaluate the histopathology sections for eligibility and locate the lesion under microscope. This initial assessment is conducted using a low-magnification in microscope. If finer structures need to be observed, the microscope is adjusted to a higher magnification. However, this manual process of analysis has several challenges such as the diagnostic results can vary due to the subjectivity of different doctors, susceptible to high error rate and increases work load of examiners (Chan, 2014). The limitation of manual examination can be addressed with automated computer aided diagnosis tool, which will assist pathologist in making accurate and precision diagnosis.

In recent years, the utilization of computer-aided diagnosis (CAD) has revolutionized the precision and efficiency of pathological examinations, particularly through image segmentation, which provides pathologists with essential evidence for their diagnostic processes (Gupta et al., 2021). In histopathology images, segmentation helps identify and highlight anomalies, aiding in accurate diagnosis. However, applying segmentation to these images poses significant challenges due to their nature, typically containing only one type of object, such as specific tissues or anomalies, which hampers the fully automatic operation of CAD-supported systems. To overcome this obstacle, CAD systems can employ deep learning models to enhance segmentation, enabling the precise detection and highlighting of anomalies (Kotadiya & Patel, 2019). As CAD technology continues to evolve, image segmentation remains a critical task that offers new opportunities in biomedical image analysis and improves the accuracy and efficiency of colorectal cancer diagnosis. Addressing the challenges of segmentation in histopathological images is essential for advancing CAD systems and supporting pathologists in making more reliable diagnoses.

Therefore, in this study we focus on the segmentation of histopathological colorectal cancer sections, encompassing six tumor differentiation regions: normal, Polyp, Low-grade intraepithelial neoplasia, High-grade intraepithelial neoplasia, Adenocarcinoma and Serrated Adenoma using the EBHI-Seg dataset Shi et al. (2023). Segmentation is performed using UNet (Ronneberger, Fischer & Brox, 2015) segmentation models intergrating with pretrained backbones VGG16 (Simonyan & Zisserman, 2014), MobileNetV1 (Howard et al., 2017), ResNet50 (He et al., 2016), and also a new generation backbone MobileViT (Mehta & Rastegari, 2021). This approach aims to compare the effectiveness of these different UNet architectures in accurately segmenting the EBHI-Seg images (Shi et al., 2023).

The main contributions of our study are as follows:

• MobileViT as a UNet Encoder: To the best of our knowledge we are the first to explore and integrate MobileViT as the encoder for the UNet architecture in the context of colorectal histopathological image segmentation.

• Comparison of different UNet backbones: We performed a comprehensive comparison of UNet models with different commonly used backbones, specifically VGG16, MobileNetV1, ResNet50, and new generation backbone MobileViT, to evaluate their performance in segmentation.

• Exploring different loss functions: In this study, we experimented with two loss functions: Binary Cross Entropy (BCE) and Dice, to assess their impact on segmentation performance. Each proposed model was trained with both loss functions to identify the optimal combination of loss function and model architecture.

• Evaluation metrics and model selection: We employed a comprehensive set of metrics, including Jacard index, Dice ratio, precision, and recall, to evaluate the experimental outcomes for each tumor stage in the test set. We have performed paired wise statistcal validations of model by conducting the Wilcoxon signed-rank test. Furthermore, to select the best model and loss combination, we utilized the TOPSIS multi-criteria decision analysis technique.

• Benchmarking against existing work: We performed an extensive comparison between our top-performing model and existing methods to demonstrate its superior segmentation capability.

Related Works

Previous studies have demonstrated the significant effectiveness of deep learning models in medical image segmentation. Rathore et al. (2019) developed a gland segmentation method for precise identification and staging of malignant colorectal cancer specimens. The approach models tissue as ellipsoids, extracting multi-scale features from glands, patches, and images. Achieving high accuracy (87.5%–88.4%) in delineating gland regions and segmenting them into component elements, the method demonstrates robust performance in segmentation metrics such as the Dice coefficient (0.87) and F-score (0.89) on the GlaS dataset. Graham et al. (2019) introduced a convolutional neural network (CNN) that addresses information loss associated with max-pooling layers by reintegrating the original image at multiple stages within the network, and employed atrous spatial pyramid pooling with diverse dilation rates to maintain resolution and facilitate multi-level aggregation. Additionally, they introduced random transformations during test time to enhance segmentation results, generating both an uncertainty map and emphasizing ambiguous regions. The proposed network achieves superior performance on the 2015 MICCAI GlaS Challenge dataset and is validated on the CRAG dataset, confirming its efficacy.

Zhao et al. (2020) presented SCAU-Net, a deep learning network inspired by the U-Net structure and attention mechanisms. With an encoder–decoder-style symmetrical architecture and the integration of spatial and channel attention modules, SCAU-Net enhances local features while suppressing irrelevant ones. Experimental evaluations on GlaS and CRAG datasets showcase its superiority over the classic U-Net, achieving a 1% enhancement in Dice score and a 1.5% improvement in Jaccard score. Rastogi, Khanna & Singh (2022) proposed a symmetric encoder–decoder network for detecting and segmenting glands in malignant subjects. Their methodology utilizes a multilevel CNN architecture to capture contextual information and integrates skip connections to concatenate features, enhancing localization accuracy. Post-processing with morphological operators further refines the predicted maps. Evaluated on the Warwick-QU dataset, their method achieves competitive results: F1-score of 0.81 for gland detection, object dice score of 0.82 for segmentation, and Hausdorff distance of 84.18 for gland shape similarity. Kassani et al. (2022) evaluated a range of deep learning architectures for automatically segmenting tumors in colorectal tissue samples. Their approach integrates convolutional neural network modules and employs transfer learning in the encoder part of the segmentation architecture for histopathology image analysis. Extensive experiments showed that the shared DenseNet and LinkNet architecture achieves state-of-the-art performance, outperforming other methods with a dice similarity index of 82.74% ± 1.77, accuracy of 87.07% ± 1.56, and an f1-score of 82.79% ± 1.79.

Wang et al. (2021) tackled the segmentation of colorectal cancer (CRC) biopsy histopathology data, specifically focusing on automating the segmentation of intraepithelial neoplasia levels. They explored two pre-processing strategies: pixel-to-propagation consistency (PPC) and Bootstrap Your Own Latent (BYOL). Their study involved initial training of the UNet coder component, demonstrating that the PPC strategy notably enhances UNet’s segmentation performance compared to BYOL. Shah et al. (2021) introduced an innovative CNN-based approach, the AtResUNet model, designed to enhance the accuracy of colorectal cancer segmentation in high-resolution medical images. This model integrates atrous convolutions, residual connections, and traditional filters. Employing an efficient patch-based method for training and inference, it reduces unnecessary computations. Trained on the DigestPath 2019 Challenge dataset, AtResUNet achieved a Dice coefficient of 0.748 for colorectal cancer segmentation. When combined with a simplified version, the ensemble achieved a higher Dice coefficient of 0.753, outperforming existing models in this domain. Dabass et al. (2023) proposed Hybrid U-Net model that incorporates advanced convolutional learning modules, attention modules, and multi-scalar transitional modules into the U-Net architecture, achieving complex multi-level convolutional feature learning. Experiments are conducted on CRAG, GlaS challenge, LC-25000, and Hospital Colon (HosC) datasets.

To the best of our knowledge from the literature, only one published study has used the Enteroscope Biopsy Histopathological Hematoxylin and Eosin Image Dataset (EBHI-Seg) for segmentation with deep learning models, specifically the study by Shi et al. (2023), on which our study is based.

Shi et al. (2023) introduced the Enteroscope Biopsy Histopathological Hematoxylin and Eosin Image Dataset (EBHI-Seg), comprising 4,456 histopathological images annotated with masks for six stages of tumor differentiation: Normal, Polyp, Intraepithelial neoplasia, Adenocarcinoma, and Serrated Adenoma. Their study compared classical machine learning methods with deep learning models like SegNet, U-Net, and MedT, highlighting superior performance of deep learning techniques. Specifically, U-Net and SegNet achieved the highest results. SegNet excelled in the normal (Dice ratio = 0.777, Jaccard index = 0.684) and low-grade intraepithelial neoplasia (Dice ratio = 0.924, Jaccard index = 0.864). Meanwhile, UNet performed well in polyp (Dice ratio = 0.965, Jaccard index = 0.308), high-grade intraepithelial neoplasia (Dice ratio = 0.895, Jaccard index = 0.816), adenocarcinoma (Dice ratio = 0.887, Jaccard index = 0.808), and Serrated Adenoma (Dice ratio = 0.938, Jaccard index = 0.886).

Methodology

Overview of proposed work

The primary goal of this study is to segment colorectal cancer at various tumor stages using U-Net models with both traditional and next-generation backbones. The study involves the following key steps:

Step 1: Preparation of the training data;

Step 2: Dataset pre-processing;

Step 3: Training deep learning segmentation models;

Step 4: Performance evaluation, comaparitive analysis and generation of outputs.

The step-by-step flow of the study is presented in Fig. 1.

Figure 1 Overview of the proposed methodology.

Dataset

In this section we discuss both the techinal and biological background of the dataset. The subsections are as follows:

Technical description

In this study, we have utilized publicly available EBHI-Seg dataset (https://doi.org/10.6084/m9.figshare.21540159.v1), comprises 4,456 histopathological images of colorectal cancer (Shi et al., 2023). This includes 2,228 histopathology section images and 2,228 corresponding ground truth images. The dataset covers five stages of tumor differentiation: Normal (76 images), Polyps (474 images), Low-grade intraepithelial neoplasia (IN) (637 images), High-grade intraepithelial neoplasia (IN) (186 images), Serrated Adenomas (58 images), Adenocarcinomas (795 images). Each histopathology section image is paired with a ground truth mask indicating the precise location of the abnormalities as determined by clinicians. The dataset, obtained through intestinal biopsy, featured images at 400× magnification using a Nissan Olympus microscope and NewUsbCamera software. The magnification details include a 10× eyepiece magnification and a 40× objective magnification.

Biological background

Normal colorectal tissue sections reveal regularly aligned tubular cells that appear unaffected characterized by structured morphology indicative of healthy tissue integrity (De Leon & Di Gregorio, 2001). In contrast, colorectal polyps, while similar in shape to normal structures, exhibit a distinct histological composition. Viewed under a microscope, polyps appear as redundant masses growing on the surfaces of body cells, often considered undesirable enlargements of the mucosal tissue (Cooper et al., 1998). Histopathological sections of polyps show intact luminal structures with minimal cellular division, primarily distinguished by a slightly higher atomic mass compared to normal tissues.

Intraepithelial neoplasia (IN) is a critical precancerous lesion characterized by increased branching of adenoid structures, dense arrangement, and varied luminal sizes compared to normal tissue. Cellular morphology shows enlarged nuclei with increased division (Ren et al., 2013). The Padova classification divides IN into low-grade and high-grade categories, with high-grade IN exhibiting more pronounced structural changes and greater nuclear enlargement than low-grade IN.

Adenocarcinoma is a malignant tumor of the digestive tract characterized by an irregular distribution of luminal structures. Identifying its boundary structures is challenging, and the nuclei are significantly enlarged at this stage (Jass & Sobin, 1989). Serrated adenomas are rare lesions, representing about 1% of all colonic polyps (Spring et al., 2006). The endoscopic appearance is not well defined but is believed to resemble that of colonic adenomas, particularly those with tubular or cerebral crypt openings (Li & Burgart, 2007). Figure 2 presents a differentiation tumor section along with the corresponding ground truth diagram.

Figure 2 Histopathological image samples of colorectal cancer with their corresponding ground truth labels.

(A) Normal tissue and ground truth, (B) Polyp and ground truth, (C) High-grade intraepithelial neoplasia and ground truth, (D) Low-grade intraepithelial neoplasia and ground truth, (E) Adenocarcinoma and ground truth, and (F) Serrated Adenoma and ground truth.

Dataset pre-processing

The preprocessing steps for this study include image resizing, data splitting, and data augmentation, normalization to prepare the dataset effectively for training the neural network.

• Image resizing: All images are resized to a consistent dimension of 224 × 224 pixels, enabling the neural network to process inputs of uniform size across different backbone models.

• Data splitting: The dataset is split into three subsets: 70% for training, 20% for validation, and 10% for testing. Specifically, the training set consisted of 1,555 images, the validation set included 446 images, and the test set comprised 255 images.

• Data augmentation: To increase data diversity and prevent overfitting, data augmentation techniques are applied exclusively to the type such as ‘Normal’, ‘High-grade IN’ and ‘Serrated Adenoma’ of the training set, which have less instances. The augmentation techniques utilized include horizontal flips, vertical flips, transpositions, and random 90-degree rotations. The N Normal class and Serrated Adenoma were augmented with all the augmentation techniques expect High-grade IN was augmented using horizontal flips, as to balance all instances of each types. These augmentations increase the variety of training samples, enhancing the model’s generalization ability. After augmentation training, the set size is 2,056.

• Image normalization: As part of the preprocessing pipeline, each pre-trained (ImageNet) U-Net backbone uses the following normalization techniques;

– MobileViT Backbone: Images are normalized by scaling pixel values to the [0, 1] range by dividing each pixel value by 255. This scaling ensures that pixel intensities are standardized for optimal performance with MobileViT.

– VGG16 Backbone: For the VGG16 backbone, pixel values are adjusted by subtracting mean RGB values of [103.939, 116.779, 123.68] (in BGR order). This normalization centers the data around a balanced mean, aligning with the expected input format of the VGG16 architecture.

– ResNet50 Backbone: Images for the ResNet50 backbone are normalized by scaling pixel values to the range [-1, 1], achieved through mean subtraction and scaling. This normalization provides a balanced input distribution suited to the ResNet50 structure.

– MobileNet Backbone: For the MobileNet backbone, pixel values are likewise normalized to the range [-1, 1] through mean subtraction and scaling. This adjustment prepares the data in a format compatible with MobileNet’s processing requirements.

Proposed segmentation models

Deep learning segmentation models are widely used in image segmentation due to their robustness in handling such tasks. Image segmentation involves dividing a digital image into various segments to extract targeted regions of interest. One category of segmentation is semantic segmentation, which entails assigning a class label to every pixel in the image. In our study, each pixel is classified into two categories background and foreground, foreground corresponds to the tumor differentiation stage in histopathology images of colorectal cancer. To achieve this, we employed U-Net (Ronneberger, Fischer & Brox, 2015) with different backbones and different loss functions, and explored its performance to meet the objectives of the study. The architecture of each backbones are discussed in following subsections.

VGG16-UNet

In this study, the VGG16-UNet model is employed for colorectal cancer segmentation, using VGG16 as the encoder to leverage its powerful feature extraction capabilities. VGG16, originally developed by Simonyan & Zisserman (2014), is well-regarded for its simplicity and effectiveness in deep learning tasks. It consists of 13 convolutional layers organized into five blocks, each followed by max-pooling layers that reduce spatial dimensions while retaining essential features. For use in segmentation, the fully connected layers of VGG16, designed for classification, are removed, allowing the architecture to integrate seamlessly as the encoder component in the VGG16-UNet model.

The decoder in VGG16-UNet closely resembles the vanilla UNet decoder, using up-sampling layers followed by convolutional layers to progressively refine the segmentation map. Skip connections are added from each encoder layer to its corresponding decoder layer, allowing high-resolution features from earlier layers to merge with the up-sampled features in the decoder, thus enhancing segmentation accuracy.

The final layer of the VGG16-UNet model consists of a single convolutional filter with a 1x1 kernel and a sigmoid activation function, producing the final segmentation output with pixel-level classification. This combination enables VGG16-UNet to benefit from VGG16’s deep feature extraction strength while maintaining the spatial detail necessary for precise segmentation through UNet’s architecture. Figure 3 illustrates the architecture of VGG16-UNet used in this study.

Figure 3 Architectural diagram of the VGG16-UNet.

ResNet50-UNet

In this study, the ResNet50-UNet model is used for colorectal cancer segmentation, incorporating the ResNet50 architecture as the encoder to leverage its advanced residual learning capabilities. Introduced by He et al. (2016), ResNet50 is well-known for its use of residual connections, which help mitigate the vanishing gradient problem and support the construction of very deep networks. ResNet50 consists of 50 layers organized into multiple residual blocks, where shortcut connections bypass one or more convolutional layers. This structure facilitates improved gradient flow and feature learning across layers. In ResNet50-UNet, the output from ResNet50’s final layer before the classification block serves as the bottleneck layer, which is then passed to the decoder. The decoder component closely follows the standard UNet design, using up-sampling layers and convolutional layers to progressively restore spatial resolution. To further enhance segmentation accuracy, skip connections are established from each encoder layer to the corresponding decoder layer, merging high-resolution features with up-sampled feature maps. The final layer of the ResNet50-UNet model includes a single convolutional filter with a 1 × 1 kernel and a sigmoid activation function, producing the segmentation map with pixel-level classification.

This combined approach allows ResNet50-UNet to capitalize on the depth and residual learning strengths of ResNet50, achieving accurate segmentation by integrating high-level semantic information with spatial detail, as facilitated by UNet’s architecture. The architecture of the ResNet50-UNet model used in this study is shown in Fig. 4.

Figure 4 Architectural diagram of the ResNet50-UNet.

MobileNet-UNet

In this study, the MobileNet-UNet model is employed for colorectal cancer segmentation, using MobileNetV1 as the encoder to maximize efficiency through depthwise separable convolutions. MobileNetV1, introduced by Howard et al. (2017), is recognized for its lightweight architecture, which significantly reduces computational complexity and memory usage while maintaining strong performance. This makes MobileNetV1 especially suitable for deployment in resource-limited environments, such as mobile devices and edge computing platforms.

The encoder in MobileNet-UNet comprises 28 layers that utilize depthwise separable convolutions. This approach splits the convolution operation into two parts: a depthwise convolution, which applies a separate filter to each input channel, and a pointwise convolution, which combines these outputs. This method greatly reduces the number of parameters and computational cost compared to standard convolutions, while still effectively capturing important features from the input images.

The decoder in the MobileNet-UNet model follows the classic UNet structure, with up-sampling and convolutional layers to gradually reconstruct the segmentation map from encoded features. Skip connections link corresponding layers of the MobileNetV1 encoder to the decoder, merging low-level spatial details with high-level semantic information. This enhances segmentation accuracy by preserving fine-grained details.

The architecture of the MobileNet-UNet model used in this study is illustrated in Fig. 5.

Figure 5 Architectural diagram of the MobileNet-UNet.

MobileViT-UNet

In this study, we propose the MobileViT-UNet model, which incorporates MobileViT as the encoder within the UNet framework to improve segmentation of colorectal cancer histopathological images. MobileViT, introduced by Mehta & Rastegari (2021), combines vision transformers with convolutional layers to leverage the strengths of both approaches, enhancing performance in image segmentation tasks.

In the MobileViT-UNet model, the encoder uses both convolutional layers and transformer blocks. The convolutional layers capture local details and texture, essential for delineating the intricate structures found in histopathological images. Meanwhile, the transformer blocks capture long-range dependencies and global context, which are particularly useful for segmenting complex and varied tumor structures. This dual approach allows the model to integrate both detailed local features and broad contextual information. The decoder in MobileViT-UNet follows the standard UNet structure, using up-sampling and convolutional layers to reconstruct the segmentation map from the encoded features. Skip connections between corresponding layers in the encoder and decoder help retain spatial details and incorporate multi-scale features, further enhancing segmentation accuracy.

By integrating MobileViT within the UNet framework, this model combines the precise feature extraction of convolutional networks with the comprehensive contextual understanding provided by transformers. Figure 6 illustrates the MobileViT-UNet architecture.

Figure 6 Architectural diagram of the MobileViT-UNet.

Training of the models

The training process involves training each model twice using two different loss functions: Binary Cross-Entropy (BCE) and Dice. This methodology resulted in two distinct instances for each model. For instance, the VGG16-UNet model one instannce is trained with the BCE loss function, resulting in the instance referred to as VGG16-UNet+BCE. It was then trained with the Dice loss function, yielding the instance named VGG16-UNet+DICE. This procedure was consistently applied across all models. This approach is to compare the effectiveness of different loss functions within the same model architecture for colorectal histopathological image segmentation.

The Binary Cross-Entropy loss function is defined as: BCE=−1N∑i=1Nyi logpi+1−yilog1−pi

where yi represents the true label, pi is the predicted probability, and N is the number of samples.

The Dice loss function is given by: Dice Loss=1−2∑i=1Nyi⋅pi+ϵ∑i=1Nyi+ ∑i=1Npi+ϵ

where yi represents the true label, pi is the predicted probability, N is the number of samples, and ϵ is a small constant added to avoid division by zero.

Other training settings are: The Adam optimizer was used consistently across all configurations, with a learning rate of 0.0001 to maintain stable training dynamics and ensure fair comparisons. This learning rate was chosen empirically after testing various values, where 0.0001 was found to provide the best convergence for most models. Each model was trained with a batch size of 8, balancing computational efficiency with convergence speed. To reduce or to avoid over fitting and enhance model generalization, early stopping was implemented by monitoring the validation loss during training. Table 1 provides a comprehensive overview of the training configurations utilized for each segmentation models

Table 1 Experimental settings for proposed model’s instances configurations (LR(α)=Learning Rate).

Model	Loss function	Optimizer	LR(α)	Epochs	Batch size	
VGG16-UNet+BCE	BCE	Adam	0.0001	14	8	
VGG16-UNet+DICE	Dice	Adam	0.0001	21	8	
ResNet50-UNet+BCE	BCE	Adam	0.0001	13	8	
ResNet50-UNet+DICE	Dice	Adam	0.0001	28	8	
MobileNet-UNet+BCE	BCE	Adam	0.0001	8	8	
MobileNet-UNet+DICE	Dice	Adam	0.0001	18	8	
MobileViT-UNet+BCE	BCE	Adam	0.0001	13	8	
MobileViT-UNet+DICE	Dice	Adam	0.0001	39	8	

Table 2 outlines the computational environment utilized for training the models. The training was conducted using Anaconda, with the TensorFlow and Keras frameworks. The hardware configuration included an NVIDIA RTX GPU with 6 GB of memory and 16 GB of RAM, providing robust computational power and ensuring efficient model training and performance.

Table 2 Configuration of the computational environment used for model training.

Component	Details	
Framework	TensorFlow and Keras	
GPU	NVIDIA RTX 3060 (6 GB)	
RAM	16 GB	
Operating System	Windows 11	
Python Version	Python 3.10	

Results and Discussion

This study investigates the segmentation of colorectal cancer tumor stages using proposed segmentation models. Each model was evaluated for its performance in distinguishing between six tumor stages: Normal, Polyp, Low-grade intraepithelial neoplasia, High-grade intraepithelial neoplasia, Serrated Adenoma, and Adenocarcinoma.

Performance metrics

To compare the effectiveness of the models, this study uses four commonly employed evaluation metrics for image segmentation tasks: the Dice ratio, Jaccard index, precision and recall. Dice ratio is a standard metric in medical imaging, frequently utilized to assess the performance of segmentation algorithms. This validation method relies on spatial overlap statistics to measure the similarity between the algorithm’s output and the ground truth. The Dice ratio is defined as: (1) Dice ratioA,B=2|A∩B||A|+|B|

where A and B represent binary masks for the segmented region and the ground truth, respectively. Dice ratio indicates the overlapping ratio between the segmented region and the ground truth. The resulting values range from 0 to 1, with higher values indicating better performance.

The Jaccard index is a classical set similarity measure with numerous applications in image segmentation. This index evaluates the similarity of two finite sets by calculating the ratio between the intersection and the union of the segmentation results and ground truth. The Jaccard index is defined as: (2) Jaccard Index=|A∩B||A∪B|

where A and B represent binary masks for the segmented region and the ground truth, respectively. The calculated results range from 0 to 1, with higher values indicating better performance.

Precision measures the accuracy of the positive predictions made by the segmentation algorithm. It is defined as the ratio of true positive pixels to the sum of true positive and false positive pixels. In other words, precision quantifies how many of the pixels predicted as part of the segmented region are actually part of the ground truth. Precision is defined as: (3) Precision=TPTP+FP

where: TP (true positives): The number of pixels correctly identified as part of the segmented region; FP (false positives): The number of pixels incorrectly identified as part of the segmented region.

Recall, also known as sensitivity or true positive rate, measures the completeness of the positive predictions. It is defined as the ratio of true positive pixels to the sum of true positive and false negative pixels. Recall quantifies how many of the actual pixels in the ground truth are correctly identified by the segmentation algorithm.Recall is defined as: (4) Recall=TPTP+FN

where: TP (true positives): The number of pixels correctly identified as part of the segmented region. FN (false negatives): The number of pixels in the ground truth that were not identified by the algorithm.

These metrics are particularly relevant to medical imaging segmentation, where accuracy and reliability are important. The Dice ratio and Jaccard index are well-suited for medical applications as they account for variations in object size and shape, providing robust measurements of spatial overlap between the segmented region and the ground truth. These metrics penalize both over- and under-segmentation, ensuring a balanced assessment of segmentation quality, which is critical when precise boundaries are needed for diagnosis. Precision measures the specificity of the segmentation, reflecting how accurately the model identifies positive regions and minimizing false positives. Recall measures sensitivity, indicating how effectively the model captures all true positive pixels, reducing the risk of missing clinically significant areas. Together, these metrics provide a comprehensive evaluation by balancing overlap accuracy with specificity and completeness, all essential for assessing performance in medical image segmentation.

Quantitative results

Table 3 provides a comprehensive analysis of the proposed segmentation model performance across various classes, utilizing key metrics such as Dice ratio, Jaccard index, precision, and recall. These metrics are pivotal in evaluating the accuracy and reliability of segmentation algorithms, crucial for applications in medical image analysis.

Table 3 Quantitative results of the proposed methods in terms of Dice ratio, Jaccard index, precision, and recall for all classes (mean ± std).

Proposed methods	Classes	Dice ratio	Jaccard index	Precision	Recall	
VGG16-UNet+BCE	Adenocarcinoma	0.887  ± 0.088	0.808  ± 0.126	0.909  ± 0.116	0.885  ± 0.101	
VGG16-UNet+Dice	0.894  ± 0.092	0.819  ± 0.128	0.902  ± 0.121	0.906  ± 0.097	
ResNet50-UNet+BCE	0.877  ± 0.085	0.79  ± 0.119	0.905  ± 0.119	0.868  ± 0.096	
ResNet50-UNet+Dice	0.884  ± 0.094	0.803  ± 0.13	0.894  ± 0.136	0.895  ± 0.087	
MobileNet-UNet+BCE	0.884  ± 0.093	0.802  ± 0.13	0.918  ± 0.108	0.869  ± 0.112	
MobileNet-UNet+Dice	0.889  ± 0.091	0.811  ± 0.129	0.906  ± 0.119	0.891  ± 0.099	
MobileViT-UNet+BCE	0.902  ± 0.079	0.829  ± 0.118	0.924  ± 0.103	0.898  ± 0.106	
MobileViT-UNet+Dice	0.925  ± 0.056	0.864  ± 0.088	0.928  ± 0.091	0.929  ± 0.054	
VGG16-UNet+BCE	High-grade IN	0.91  ± 0.058	0.839  ± 0.09	0.902  ± 0.101	0.928  ± 0.054	
VGG16-UNet+Dice	0.916  ± 0.068	0.851  ± 0.103	0.894  ± 0.113	0.95  ± 0.043	
ResNet50-UNet+BCE	0.912  ± 0.06	0.843  ± 0.091	0.91  ± 0.109	0.923  ± 0.039	
ResNet50-UNet+Dice	0.913  ± 0.064	0.845  ± 0.096	0.906  ± 0.113	0.93  ± 0.028	
MobileNet-UNet+BCE	0.907  ± 0.052	0.834  ± 0.081	0.916  ± 0.1	0.909  ± 0.049	
MobileNet-UNet+Dice	0.916  ± 0.063	0.85  ± 0.096	0.906  ± 0.111	0.938  ± 0.038	
MobileViT-UNet+BCE	0.923  ± 0.05	0.861  ± 0.08	0.912  ± 0.094	0.943  ± 0.038	
MobileViT-UNet+Dice	0.928  ± 0.045	0.869  ± 0.073	0.915  ± 0.083	0.948  ± 0.023	
VGG16-UNet+BCE	Low-grade IN	0.96  ± 0.015	0.924  ± 0.028	0.974  ± 0.02	0.947  ± 0.024	
VGG16-UNet+Dice	0.963  ± 0.015	0.929  ± 0.027	0.97  ± 0.023	0.956  ± 0.02	
ResNet50-UNet+BCE	0.952  ± 0.02	0.909  ± 0.036	0.974  ± 0.025	0.932  ± 0.033	
ResNet50-UNet+Dice	0.959  ± 0.017	0.922  ± 0.032	0.97  ± 0.029	0.949  ± 0.023	
MobileNet-UNet+BCE	0.947  ± 0.029	0.9  ± 0.05	0.98  ± 0.019	0.917  ± 0.051	
MobileNet-UNet+Dice	0.958  ± 0.021	0.92  ± 0.037	0.97  ± 0.03	0.947  ± 0.03	
MobileViT-UNet+BCE	0.962  ± 0.018	0.927  ± 0.032	0.977  ± 0.02	0.948  ± 0.028	
MobileViT-UNet+Dice	0.964  ± 0.013	0.931  ± 0.024	0.979  ± 0.02	0.95  ± 0.02	
VGG16-UNet+BCE	Normal	0.956  ± 0.013	0.916  ± 0.024	0.97  ± 0.035	0.944  ± 0.019	
VGG16-UNet+Dice	0.958  ± 0.014	0.919  ± 0.025	0.971  ± 0.034	0.946  ± 0.021	
ResNet50-UNet+BCE	0.949  ± 0.018	0.904  ± 0.032	0.971  ± 0.041	0.931  ± 0.022	
ResNet50-UNet+Dice	0.95  ± 0.022	0.905  ± 0.038	0.965  ± 0.05	0.937  ± 0.019	
MobileNet-UNet+BCE	0.955  ± 0.013	0.914  ± 0.024	0.974  ± 0.028	0.938  ± 0.02	
MobileNet-UNet+Dice	0.956  ± 0.015	0.917  ± 0.027	0.97  ± 0.028	0.943  ± 0.017	
MobileViT-UNet+BCE	0.959  ± 0.012	0.922  ± 0.022	0.983  ± 0.01	0.937  ± 0.025	
MobileViT-UNet+Dice	0.957  ± 0.014	0.918  ± 0.025	0.988  ± 0.008	0.928  ± 0.026	
VGG16-UNet+BCE	Polyp	0.957  ± 0.016	0.917  ± 0.028	0.961  ± 0.029	0.952  ± 0.016	
VGG16-UNet+Dice	0.957  ± 0.014	0.919  ± 0.026	0.967  ± 0.027	0.949  ± 0.013	
ResNet50-UNet+BCE	0.955  ± 0.014	0.915  ± 0.026	0.974  ± 0.018	0.938  ± 0.02	
ResNet50-UNet+Dice	0.957  ± 0.014	0.917  ± 0.025	0.97  ± 0.023	0.944  ± 0.016	
MobileNet-UNet+BCE	0.955  ± 0.019	0.914  ± 0.034	0.977  ± 0.017	0.934  ± 0.03	
MobileNet-UNet+Dice	0.959  ± 0.013	0.922  ± 0.024	0.97  ± 0.022	0.949  ± 0.013	
MobileViT-UNet+BCE	0.959  ± 0.013	0.921  ± 0.023	0.968  ± 0.023	0.95  ± 0.017	
MobileViT-UNet+Dice	0.957  ± 0.013	0.917  ± 0.023	0.975  ± 0.019	0.94  ± 0.019	
VGG16-UNet+BCE	Serrated adenoma	0.951  ± 0.019	0.908  ± 0.035	0.96  ± 0.026	0.943  ± 0.027	
VGG16-UNet+Dice	0.946  ± 0.025	0.899  ± 0.045	0.956  ± 0.031	0.938  ± 0.037	
ResNet50-UNet+BCE	0.931  ± 0.03	0.872  ± 0.052	0.957  ± 0.032	0.908  ± 0.049	
ResNet50-UNet+Dice	0.947  ± 0.022	0.9  ± 0.04	0.945  ± 0.037	0.949  ± 0.027	
MobileNet-UNet+BCE	0.946  ± 0.014	0.899  ± 0.024	0.974  ± 0.021	0.921  ± 0.031	
MobileNet-UNet+Dice	0.945  ± 0.018	0.895  ± 0.032	0.959  ± 0.027	0.932  ± 0.036	
MobileViT-UNet+BCE	0.96  ± 0.009	0.923  ± 0.017	0.965  ± 0.023	0.955  ± 0.013	
MobileViT-UNet+Dice	0.953  ± 0.017	0.911  ± 0.031	0.957  ± 0.032	0.951  ± 0.027	
VGG16-UNet+BCE	All region (average)	0.937  ± 0.035	0.885  ± 0.055	0.946  ± 0.054	0.933  ± 0.04	
VGG16-UNet+Dice	0.939  ± 0.038	0.889  ± 0.059	0.943  ± 0.058	0.941  ± 0.038	
ResNet50-UNet+BCE	0.929  ± 0.038	0.872  ± 0.059	0.949  ± 0.057	0.917  ± 0.043	
ResNet50-UNet+Dice	0.935  ± 0.039	0.882  ± 0.06	0.942  ± 0.065	0.934  ± 0.033	
MobileNet-UNet+BCE	0.932  ± 0.037	0.877  ± 0.057	0.957  ± 0.049	0.915  ± 0.049	
MobileNet-UNet+Dice	0.937  ± 0.037	0.886  ± 0.057	0.947  ± 0.056	0.933  ± 0.039	
MobileViT-UNet+BCE	0.944  ± 0.03	0.897  ± 0.049	0.955  ± 0.046	0.939  ± 0.038	
MobileViT-UNet+Dice	0.947  ± 0.026	0.902  ± 0.044	0.957  ± 0.042	0.941  ± 0.028	
Notes.

The best results are shown in bold.

From Table 3, we can observe that the averaged performance across all regions highlights MobileViT-UNet+Dice as the leading model, with the highest values across key metrics: a Dice Ratio of 0.947 ± 0.026, Jaccard index of 0.902 ± 0.044, precision of 0.957 ± 0.042, and recall of 0.941 ± 0.028. These results underscore its superior capability in boundary delineation and segmentation accuracy, minimizing false positives and negatives. Other models also demonstrated strong performance in segmenting across regions, though with minor variations. The MobileViT-UNet with Binary Cross-Entropy (BCE) loss, for instance, performed comparably well, achieving a Dice Ratio of 0.944 ± 0.030 and a Jaccard index of 0.897 ± 0.049, with precision at 0.955 ± 0.046 and recall at 0.939 ± 0.038. This model maintains a high balance between segmentation accuracy and precision but is outperformed by its Dice-loss counterpart. MobileNet-UNet, VGG16-UNet and ResNet50-UNet models also show substantial performance with both BCE and Dice loss, but with lower performance consistency compared to MobileViT-UNet in handling all-region segmentation. Therefore quantitative analysis supports MobileViT-UNet+Dice as the optimal model for segmentation tasks across all tissue types.

Statistical analysis

The models’ effectiveness and robustness are evaluated using a non-parametric statistical test. MobileViT-UNet+Dice, identified as the top-performing model from Table 3, is compared pairwise against other methods using the Wilcoxon signed rank test (WSRT) with significance criteria p ≤ 0.05 as significant. The WSRT was applied on a per-image basis for each model pair across evaluation metrics—Dice ratio, Jaccard index, precision, and recall. This per-image approach captures variability across individual images, providing a more detailed assessment of model performance differences. The models per image performance across all metrics is provided as box plot in Figs. 7, 8, 9 and 10, along with annotation of pairwise WSRT significance.

Figure 7 The box plot (methods versus Dice ratio) of comprehensive performance of methods over test set images along with pairwise statistical test p-value annotations, where annotation are - ns: Not significant, *: 1 × 10−2 < p ≤ 5.00 × 10−2, **:1 × 10−3 < p ≤ 1 × 10−2, ***:1 × 10−4 < p ≤ 1 × 10−3, ****:p ≤ 1 × 10−4.

Figure 8 The box plot (methods versus Jaccard index) of comprehensive performance of methods over test set images along with pairwise statistical test p-value annotations, where annotation are - ns: Not significant, *:1 × 10−2 < p ≤ 5.00 × 10−2, **:1 × 10−3 < p ≤ 1 × 10−2, ***:1 × 10−4 < p ≤ 1 × 10−3, ****:p ≤ 1 × 10−4.

Figure 9 The box plot (methods versus precision) of comprehensive performance of methods over test set images along with pairwise statistical test p-value annotations, where annotation are - ns: Not significant, *:1 × 10−2 < p ≤ 5.00 × 10−2, **:1 × 10−3 < p ≤ 1 × 10−2, ***:1 × 10−4 < p ≤ 1 × 10−3, ****:p ≤ 1 × 10−4.

Figure 10 The box plot (methods versus precision) of comprehensive performance of methods over test set images along with pairwise statistical test p-value annotations, where annotation are - ns: Not significant, *:1 × 10−2 < p ≤ 5.00 × 10−2, **:1 × 10−3 < p ≤ 1 × 10−2, ***:1 × 10−4 < p ≤ 1 × 10−3, ****:p ≤ 1 × 10−4.

The WSRT results for each metric are shown in Tables 4, 5, 6 and 7.

From Table 4, we observe that MobileViT-UNet+Dice performs significantly better on the Dice ratio than other models, with particularly comparable performance to MobileViT-UNet+BCE, where the results indicate similar efficacy. Table 5 highlights MobileViT-UNet+Dice’s strong results for the Jaccard index as well, showing significantly better performance than other models, further indicating its robustness in this metric.

In Table 6, MobileViT-UNet+Dice maintains high precision, significantly outperforming nearly all models except for MobileViT-UNet+BCE, with which it shares non-significant results, suggesting comparable precision performance. Likewise, in Table 7, MobileViT-UNet+Dice demonstrates strong recall scores, with significant differences in favor of this model over most alternatives. Notable exceptions are the comparisons with VGG16-UNet+Dice and MobileNet-UNet+Dice, where performance differences are non-significant.

Therefore the WSRT results across all metrics consistently reinforce MobileViT-UNet+Dice as a highly effective and robust model, sustaining superior or comparable performance relative to other models across each evaluation metric.

Multi-criteria decision analysis for ranking segmentation models

From the statistical analysis we observed some methods have no significant difference in performance across some evaluation metrics therefore it is difficult to select the best one. In order to identify the most best performing model, we conducted a multi-criteria decision analysis (MCDA). MCDA is a systematic approach for evaluating and ranking based on multiple criteria in complex decision-making scenarios. This method facilitates informed decision-making by balancing and considering multiple perspectives.

Table 4 Wilcoxon test P-values (paired samples) of Dice ratio values of all methods.

Comparison	P-value	Significance	
MobileViT-UNet+Dice vs. VGG16-UNet+Dice	1.45E−04	Significant	
MobileViT-UNet+Dice vs. VGG16-UNet+BCE	1.59E−10	Significant	
MobileViT-UNet+Dice vs. ResNet50-UNet+Dice	8.68E−11	Significant	
MobileViT-UNet+Dice vs. ResNet50-UNet+BCE	1.20E−23	Significant	
MobileViT-UNet+Dice vs. MobileNet-UNet+BCE	1.92E−17	Significant	
MobileViT-UNet+Dice vs. MobileNet-UNet+Dice	2.00E−06	Significant	
MobileViT-UNet+Dice vs. MobileViT-UNet+BCE	1.77E−01	Non-Significant	

Table 5 Wilcoxon test P-values (paired samples) of Jaccard index values of all methods.

Comparison	P-value	Significance	
MobileViT-UNet+Dice vs. VGG16-UNet+Dice	1.46E−04	Significant	
MobileViT-UNet+Dice vs. VGG16-UNet+BCE	1.33E−10	Significant	
MobileViT-UNet+Dice vs. ResNet50-UNet+Dice	7.53E−11	Significant	
MobileViT-UNet+Dice vs. ResNet50-UNet+BCE	9.15E−24	Significant	
MobileViT-UNet+Dice vs. MobileNet-UNet+BCE	1.68E−17	Significant	
MobileViT-UNet+Dice vs. MobileNet-UNet+Dice	1.92E−06	Significant	
MobileViT-UNet+Dice vs. MobileViT-UNet+BCE	1.91E−01	Non-Significant	

Table 6 Wilcoxon test P-values (paired samples) of precision values of all methods.

Comparison	P-value	Significance	
MobileViT-UNet+Dice vs. MobileViT-UNet+BCE	3.296E−04	Significant	
MobileViT-UNet+Dice vs.. MobileNet-UNet+Dice	4.735E−06	Significant	
MobileViT-UNet+Dice vs. MobileNet-UNet+BCE	6.786E−01	Non-Significant	
MobileViT-UNet+Dice vs. ResNet50-UNet+Dice	1.482E−07	Significant	
MobileViT-UNet+Dice vs. ResNet50-UNet+BCE	5.872E−04	Significant	
MobileViT-UNet+Dice vs. VGG16-UNet+Dice	3.597E−14	Significant	
MobileViT-UNet+Dice vs. VGG16-UNet+BCE	4.021E−11	Significant	

Table 7 Wilcoxon test P-values (paired samples) of recall values of all methods.

Comparison	P-value	Significance	
MobileViT-UNet+Dice vs. VGG16-UNet+Dice	1.23E−02	Significant	
MobileViT-UNet+Dice vs. VGG16-UNet+BCE	4.04E−02	Significant	
MobileViT-UNet+Dice vs. ResNet50-UNet+Dice	5.41E−03	Significant	
MobileViT-UNet+Dice vs. ResNet50-UNet+BCE	1.50E−14	Significant	
MobileViT-UNet+Dice vs. MobileNet-UNet+Dice	1.66E−01	Non-Significant	
MobileViT-UNet+Dice vs. MobileNet-UNet+BCE	4.14E−16	Significant	
MobileViT-UNet+Dice vs. MobileViT-UNet+BCE	1.16E−01	Non-Significant	

In this study, we employed an prominent MCDA technique i.e Technique for Order of Preference by Similarity to Ideal Solution (TOPSIS) combined with the entropy weighting methodology for our performance criteria. The assessment criteria that we used for assessment are Dice ratio, Jaccard index, precision, and recall. The result of TOPSIS and ranking of model is presented on Table 8. The TOPSIS scores acrross all tissue type are represent as bar graph in Fig. 11. Table 8 reveal that the MobileViT-UNet+Dice model achieved the highest TOPSIS scores across all classes therefore attaining the highest ranking among all models. MobileViT-UNet+BCE obtains the second ranking and ResNet50-UNet+BCE is ranked the lowest. This outcome underscores MobileViT-UNet+Dice superior performance in our study and supports its selection as the most effective model for segmenting colorectal regions in histopathology images.

Table 8 TOPSIS score and ranking of all methods achieved across all region.

Proposed methods	TOPSIS_Score of all regions	Rank	
MobileViT-UNet+Dice	1	1	
MobileViT-UNet+BCE	0.862043077	2	
VGG16-UNet+Dice	0.680611636	3	
MobileNet-UNet+Dice	0.547552944	4	
VGG16-UNet+BCE	0.5276324	5	
ResNet50-UNet+Dice	0.482292523	6	
MobileNet-UNet+BCE	0.164548954	7	
ResNet50-UNet+BCE	0.070319059	8	
Notes.

The best results are shown in bold.

Figure 11 Bar graph of TOPSIS score and ranking of all methods achieved across each tissue type.

Qualitative results

In addition to quantitative metrics, qualitative assessments were conducted to evaluate the visual and practical performance of each model in segmenting colorectal cancer tumor stages. Qualitative results offer insights into the model’s effectiveness from a visual standpoint and its ability to handle complex patterns in images. Figure 12 illustrate these comparisons for various models, highlighting differences in generation of segmentation mask. We observe distinct performance characteristics of each model in segmenting colorectal cancer tumor stages. The MobileViT-UNet+Dice model delivers highly accurate segmentations, with clear and well-defined tumor boundaries that closely align with the ground truth annotations. This model excels in capturing intricate details and provides precise delineation of different tumor stages of colerectal cancer. In contrast, the ResNet50-UNet+Dice model, VGG16-UNet Dice shows some limitations, including less well-defined boundaries and a tendency towards blurring.

Figure 12 Visual of segmentation mask generated by proposed models of all classes.

Benchmarking against existing work

In this study, we compare our results with a previous existing work (Shi et al., 2023). Shi et al. (2023) introduced the dataset used in our research and proposed three deep learning segmentation models: U-Net, SegNet, and MedT. Shi et al. (2023) provided a comprehensive evaluation of these models on this dataset, which serves as a robust benchmark for our comparisons.

Our analysis aims to highlight the improvements and differences in performance metrics between our best performing proposed model (MobileViT-UNet+Dice) and the models by Shi et al. (2023). Table 9 provides a detailed comparison, focusing on key metrics such as dice ratio, jaccard index, precision, and recall.

Table 9 Comparison of the proposed best performing model (MobileViT-UNet+Dice) with existing models in terms of Dice ratio, Jaccard index, precision, and recall for all classes.

Methods	Classes	Dice Ratio	Jaccard Index	Precision	Recall	
Adenocarcinoma	UNet	0.887	0.808	0.85	0.95	
Seg-Net	0.865	0.775	0.792	0.977	
MedT	0.735	0.595	0.662	0.864	
Proposed(Best)	0.925	0.864	0.928	0.929	
High-grade IN	UNet	0.895	0.816	0.847	0.961	
Seg-Net	0.894	0.812	0.881	0.913	
MedT	0.824	0.707	0.74	0.958	
Proposed(Best)	0.928	0.869	0.915	0.948	
Low-grade IN	UNet	0.911	0.849	0.879	0.953	
Seg-Net	0.924	0.864	0.883	0.977	
MedT	0.889	0.808	0.876	0.916	
Proposed(Best)	0.964	0.931	0.979	0.95	
Normal	UNet	0.411	0.263	0.586	0.328	
Seg-Net	0.777	0.684	0.895	0.758	
MedT	0.676	0.562	0.874	0.61	
Proposed(Best)	0.957	0.918	0.988	0.928	
Polyp	UNet	0.965	0.308	0.496	0.47	
Seg-Net	0.937	0.886	0.916	0.965	
MedT	0.771	0.643	0.687	0.92	
Proposed(Best)	0.957	0.917	0.975	0.94	
Serrated Adenoma	UNet	0.938	0.886	0.899	0.983	
Seg-Net	0.907	0.832	0.859	0.963	
MedT	0.67	0.509	0.896	0.544	
Proposed(Best)	0.953	0.911	0.957	0.951	
All region	UNet	0.835	0.655	0.76	0.774	
Seg-Net	0.884	0.809	0.871	0.926	
MedT	0.761	0.637	0.789	0.802	
Proposed(Best)	0.947	0.902	0.957	0.941	
Notes.

The best results are shown in bold.

Based on Table 9, the results indicate that our best proposed model (MobileViT-UNet+Dice) consistently surpasses the performance of existing models(U-Net, Seg-Net, and MedT) introduced by Shi et al. (2023) across various classes and metrics. For the class Normal, our model achieves a Dice score of 0.957, a Jaccard index of 0.918, precision of 0.988, and recall of 0.928, significantly outperforming the other models. Similarly, for the Polyp class, our model achieves a high Dice score of 0.957, Jaccard index of 0.917, and Precision of 0.975; although it slightly trails Seg-Net in recall, the overall performance remains superior.

In the High-grade IN class, our model demonstrates a clear advantage with a Dice score of 0.928 and high precision of 0.915, underscoring its accuracy in segmenting this category. For the Low-grade IN class, our model achieves the highest Dice score of 0.964 and maintains consistently high values across all metrics, underscoring its robustness. In the Adenocarcinoma class, our model leads with a Dice score of 0.925, indicating its effectiveness in this challenging category.

For Serrated Adenoma, our model excels with a Dice score of 0.953 and shows superior performance in Jaccard index, precision, and recall, outperforming the other models in all metrics for this class. Considering the average across all regions, our model achieves the highest values, with a Dice score of 0.947, Jaccard index of 0.902, precision of 0.957, and recall of 0.941, illustrating its comprehensive efficacy in segmentation. Overall, these results underscore the superiority of our proposed model across all metrics compared to existing models

Limitation and validity of the study

This study focused on segmenting the differentiation stages of colorectal cancer. Despite the promising results, a key limitation noted was that the dataset used was limited and lacked diversity. This constraint hindered the assessment of the generalization ability of the trained models. While rigorous assessments were conducted to identify the best-performing model on this dataset, the study does not provide insights into model performance in out-of-distribution settings due to the limited data. Therefore, future work could explore using a larger and more diverse dataset, ideally sourced from multiple institutions, to enhance the robustness and generalization of the trained models in multi-domain scenarios and also explore foundational models embeddings on diverse datasets.

Despite these limitations, the study maintains strong internal validity through the use of robust methodologies, including data augmentation and early stopping. Additionally, validity is enhanced through the use of standardized evaluation metrics and a variety of model architectures, which supports the generalizability of the findings to similar settings. The application of MCDA, specifically the TOPSIS technique with information entropy weighting, provides a rigorous framework for comparing and ranking model performance across multiple criteria, further enhancing the study’s validity.

Reproducibility

Ensuring the reproducibility of a study is crucial for validating its findings and facilitating future research. Comprehensive details of the computing infrastructure used in this study are provided in Table 2. To support reproducibility, the weights of the best-performing trained model, datasets information, and codes are made publicly available on: https://doi.org/10.5281/zenodo.14048544. Additionally, detailed documentation offers instructions on setting up the computing environment, running the code, and interpreting the results.

Conclusion

In this study, we thoroughly investigated the performance of four segmentation models—VGG16-UNet, ResNet50-UNet, MobileNetV1-UNet, and MobileViT-UNet—in segmenting six colorectal cancer regions-normal, polyp, high-grade IN, low-grade IN, adenocarcinoma, and serrated adenoma. In this study, we have made a first endeavour to integrate new generation computer vision model MobileViT as backbone of UNet. MobileViT-UNet model effectively captures both fine-grained spatial information and long-range dependencies, leading to superior segmentation performance. Each segmentation models was trained with two distinct loss functions, binary cross-entropy and dice loss, and underwent comprehensive evaluation using critical metrics such as Dice ratio, Jaccard index, precision, and recall. The results demonstrated that the MobileViT-UNet+Dice model, consistently outperformed the other models across all evaluation metrics.

The MobileViT-UNet+Dice model achieved Dice ratio of 0.947 ± 0.026, Jaccard Index of 0.902 ± 0.044, precision of 0.957 ± 0.042, and recall of 0.941 ± 0.028. across all colorectal cancer classes. These metrics underscore the model’s ability to accurately delineate cancerous regions, significantly enhancing the precision of colorectal cancer segmentation. We also performed pairwise WSRT across metrics—Dice ratio, Jaccard index, precision, and recall to further validate MobileViT-UNet+Dice performance compared to other model. From the analysis result it is established that MobileViT-UNet+Dice outperforms other models. Further to to obtain the best performing segmentation based on multiple criteria we employed an MCDA technique, namely TOPSIS, with entropy weighting. The analysis also revealed that the MobileViT-UNet+Dice model achieved the highest TOPSIS scores across all classes, solidifying its ranking as the best-performing model.

This study also benchmarks against existing work proposed in Shi et al. (2023). The models compared are UNet, Seg-Net, and MedT. We conducted a comparative analysis using our best-performing model, MobileViT-UNet+Dice. The results demonstrate that MobileViT-UNet+Dice achieves higher metric scores across all classes compared to the existing methods, highlighting its superiority over all the other models.

Future work could explore further refinements to these models, including experimenting with different transformer architectures, additional data augmentation techniques, incorporating larger multi-institutional datasets and extending the application of these models to other types of medical imaging. This continued research could further validate the generalizability and robustness of our approach, ultimately contributing to the advancement of computer-aided diagnostic systems in medical imaging.

Additional Information and Declarations

Competing Interests

Author Contributions

Data Availability

The authors declare there are no competing interests.

Barun Barua conceived and designed the experiments, performed the experiments, analyzed the data, performed the computation work, prepared figures and/or tables, and approved the final draft.

Genevieve Chyrmang conceived and designed the experiments, performed the experiments, analyzed the data, performed the computation work, prepared figures and/or tables, and approved the final draft.

Kangkana Bora conceived and designed the experiments, performed the experiments, performed the computation work, prepared figures and/or tables, authored or reviewed drafts of the article, and approved the final draft.

Manob Jyoti Saikia conceived and designed the experiments, analyzed the data, authored or reviewed drafts of the article, and approved the final draft.

The following information was supplied regarding data availability:

The weights of the best performing trained model, datasets information, and codes are made publicly available at Zenodo: Barua, B. (2024). Optimizing Colorectal Cancer Segmentation with MobileViT-UNet and Multi-Criteria Decision Analysis. Zenodo. https://doi.org/10.5281/zenodo.14048544.

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
