# Peer review of "Optimizing colorectal cancer segmentation with MobileViT-UNet and multi-criteria decision analysis"

_PeerJ Computer Science, doi:10.7717/peerj-cs.2633_

## Round 0.1 · original submission · Major Revisions

Dear authors,

You are advised to critically respond to all comments point by point when preparing an updated version of the manuscript and while preparing for the rebuttal letter. Please address all comments/suggestions provided by reviewers, considering that these should be added to the new version of the manuscript.

Kind regards,
PCoelho

Reviewer 1 ·

Basic reporting

The authors propose the development of computer-aided diagnostic (CAD) systems using various UNet architectures integrated with different backbone models. The improvement of segmentation accuracy in histopathological images is the main focus. A comparison between different models has been made.

- The manuscript may benefit from improved clarity and professional English language usage. Certain phrases could be restructured for better comprehension, particularly in the introduction and methodology sections, where complex ideas are presented. i.e. "Manual examination methods suffer from limitations such as subjectivity and data overload".

Experimental design

Data Preprocessing Discussion: Add detailed discussion regarding data preprocessing steps - very important. How the data was prepared for the models?

Validity of the findings

- Evaluation Metrics Justification: Various evaluation metrics are used. However, a deeper discussion on why certain metrics are more suitable for this study. In addition, providing more robust statistical validation of the results would support the claims made regarding model performance and effectiveness.

Additional comments

In the conclusions, the authors can add limitations of this study and suggest future research directions. Addressing unresolved questions or challenges encountered during the research would provide a more balanced view and guide subsequent studies in this area.

Reviewer 2 ·

Basic reporting

The manuscript by Barua et al presents a comparison of deep learning based architectures for segmenting colorectal cancer (CRC) tissue into specific classes representing normal tissue and various forms of cancer tissue. The compared methods cover variety of U-Net variants. Public CRC dataset with ground truth segmentations for the pre-defined tissue types is used.

The article is written in clear English, with some types here and there (please check carefully). Background includes fairly satisfactory coverage of relevant literature, however, recent developments in foundation models are not included and should be added for completeness, and preferably also for comparison. Format of the paper is logical, except the dataset description, which is a mix of biomedical background of the tissue types and the dataset description - the former would better fit to background.

Experimental design

Methods are described satisfactorily in general. Experimental design needs to be clarified. Please state clearly whether you divide the dataset into training, test and validation sets before augmentation - this is absolutely critical as division randomly after augmentation will lead to significant positive bias as augmented versions of the samples are likely to be found both from training and test/validation sets.

In presenting the results, it would be good to give the deviation +/- in addition to the mean of the metrics.

As mentioned, it would be great to include state-of-the-art foundation model for comparison, the least would be to discuss it.

The experimental setup includes parameter selection for various training related hyperparameters - how are these values determined? I hope and expect that optimizing these values has not been done through repeatedly running the validation set.

Validity of the findings

With the assumption that the train/test/validation has been divided such that no leakage between categories happens, the findings appear valid.

---

## Round 0.2 · accepted · Accept

Dear authors, we are pleased to verify that you meet the reviewer's valuable feedback to improve your research.

Getting feedback from any of the reviewers from previous rounds was not possible, as the editor, I carried out the review. After verification, the manuscript is ready to be accepted.

Thank you for considering PeerJ Computer Science and submitting your work.